# Automatic Classification of Myocardial Infarction Using Spline Representation of Single-Lead Derived Vectorcardiography

**DOI:** 10.3390/s20247246

**Published:** 2020-12-17

**Authors:** Yu-Hung Chuang, Chia-Ling Huang, Wen-Whei Chang, Jen-Tzung Chien

**Affiliations:** Institute of Electrical and Computer Engineering, National Chiao-Tung University, Hsinchu 30010, Taiwan; yuhung1206.eed09g@nctu.edu.tw (Y.-H.C.); alin0624.eed05@nctu.edu.tw (C.-L.H.); jtchien@nctu.edu.tw (J.-T.C.)

**Keywords:** electrocardiography, vectorcardiography, myocardial infarction, long short-term memory, spline, multilayer perceptron

## Abstract

Myocardial infarction (MI) is one of the most prevalent cardiovascular diseases worldwide and most patients suffer from MI without awareness. Therefore, early diagnosis and timely treatment are crucial to guarantee the life safety of MI patients. Most wearable monitoring devices only provide single-lead electrocardiography (ECG), which represents a major limitation for their applicability in diagnosis of MI. Incorporating the derived vectorcardiography (VCG) techniques can help monitor the three-dimensional electrical activities of human hearts. This study presents a patient-specific reconstruction method based on long short-term memory (LSTM) network to exploit both intra- and inter-lead correlations of ECG signals. MI-induced changes in the morphological and temporal wave features are extracted from the derived VCG using spline approximation. After the feature extraction, a classifier based on multilayer perceptron network is used for MI classification. Experiments on PTB diagnostic database demonstrate that the proposed system achieved satisfactory performance to differentiating MI patients from healthy subjects and to localizing the infarcted area.

## 1. Introduction

Myocardial infarction (MI) has long been recognized as the main cause of death worldwide. According to the data from the World Health Organization (WHO) [1], cardiovascular diseases, including MI, were estimated to account for 31% of deaths worldwide in 2017. In the United States, about 110,000 Americans died of MI in 2015 and the estimated annual incidence of MI is 605,000 new attacks [2]. MI results from an occlusion of the coronary artery and insufficient blood supply to the myocardium. It can be further classified into various subtypes depending on the localization of infarcted area. In clinical setting, MI is diagnosed using 12-lead electrocardiography (ECG) [3] as well as 3-lead vectorcardiography (VCG) [4]. ECG signals are recorded from different locations of the body to capture the three-dimensional view of the human heart. The standard ECG has 12 leads, including six limb leads (I, II, III, aVR, aVL, aVF) and six chest leads ($V_{1}$ to $V_{6}$). Figure 1 shows the three-dimensional view of 12 standard leads on the xyz-coordinate axis system. According to electrode positioning, the 12 ECG leads can be used to localize different types of MI, such as inferior leads (II, III, aVF), septal leads $\left(V_{1},V_{2}\right)$, anterior leads $\left(V_{3},V_{4}\right)$, and lateral leads (I, aVL, $V_{5}$, $V_{6}$). A typical waveform of the ECG beat consists of a P wave, a QRS-complex, and a T wave. These characteristic waves correspond to the sequence of depolarization and repolarization of the atria and ventricles. ECG signs suggestive of MI include ST-segment deviation or changes in the shapes of Q-wave and T-wave, using which physicians can localize damage to specific areas of the heart. However, it may be noted that 12-lead ECG requires ten electrodes for recording and some of the leads contain redundant information. Instead, VCG requires a minimum of four electrodes and it monitors cardiac electrical activity in three orthogonal planes of the body [5]. Generally, Frank leads $\left(V_{x},V_{y},V_{z}\right)$ scanned in orthogonal xyz axes are used for VCG measurements. The main advantage of VCG is that it uses fewer leads than 12-lead ECG for medical diagnostic applications. Moreover, different studies [6,7,8] have demonstrated that VCG provides a higher sensitivity for the diagnosis of MI as well as ischemic heart diseases. In this study, VCG signal is processed to extract clinically significant features that will allow for MI classification.

MI is also known as a silent heart attack that usually occurs without clear symptoms. Hence, early diagnosis and timely treatment are crucial to improve the recovery rate of MI patients. In recent years, several computer-aided diagnostic methods have been proposed for automatic MI detection and localization [9,10,11,12,13,14,15,16,17,18]. Most of these approaches extract the clinically significant features from the ECG signal and then apply an appropriate classifier in the classification stage. Various informative features have been extracted to represent the ECG beats, such as morphological features [10] as well as frequency and wavelet-based features [11,12]. Moreover, some studies have attempted to use directly measured or derived VCG to identify changes in the VCG morphology such as the QRS and T-wave loops [16,17,18]. For classification, different machine learning algorithms have been investigated, including k-nearest neighbors (KNN) [10,12], artificial neural network (ANN) [11], recurrent neural network (RNN) [13] and convolutional neural network (CNN) [14,15]. Furthermore, several researchers [13,14,15] have proposed end-to-end approaches for MI detection and localization. These methods obviate the need to extract features at the cost of higher computational complexity. ECG abnormalities due to MI may be observed in the ST-segment deviation or changes in the shapes of T-wave and Q-wave. Generally, it is a prerequisite to identify characteristic waves of ECG beats before performing the feature extraction. Although various methods have been proposed for ECG wave delineation [19,20,21,22], they still have some limitations for characterization of MI beats. To address this constraint, we apply spline curve fitting [23,24] to the entire heartbeat to model all of the characteristic waves and use fitted coefficients as features. The advantage of using the entire heartbeat is that the QRS complexes and P and T waves can be included in the curve fitting so that poor quality features resulting from delineation errors can be avoided. Moreover, the VCG signal is semiperiodic in nature and has numerous clinically relevant turning points in each heartbeat. Such signals require a higher-order polynomial to fit, leading to severe oscillations of the fitted curve which cause the overfitting problem [25]. By contrast, the spline’s flexibility in approximating curves with different degrees of smoothness at different locations is ideal for representing the semiperiodic VCG signal.

Another problem which requires further investigation is to test the feasibility of single-lead ECG in classifying different types of MI. Several wearable devices which use single-lead ECG to facilitate continuous ambulatory monitoring have recently appeared on the market [26]. While these devices make regular ECG recording possible, their practical applicability for cardiac diagnostics remain limited. This is because physicians need checking ECG patterns to diagnose by correlating information from two or more ECG leads. For example, abnormalities in chest leads ($V_{1}$ to $V_{4}$) are suggestive of a problem in the posterior wall of the heart and no abnormalities will be detected by a single lead [27]. The ability to transform from single-lead ECG to 12-lead ECG enables the wider use of wearable devices for clinical diagnostic applications. However, prior attempts to synthesize 12-lead ECG or 3-lead VCG from a single lead have not been successful. Most existing lead transformation approaches require at least two synchronously acquired leads [28,29,30,31,32,33,34,35,36,37,38,39,40], hampering their applicability to the present context. This has motivated our investigation into trying to synthesize the 3-lead VCG from single-lead ECG signal. Since lead I is provided by most wearable devices, we propose a derived VCG system by considering the lead I ECG signal as input and three Frank leads as output of the system.

A lot of emphases have been recently put on derived ECG systems due to the increasing demand of personalized healthcare applications. The methods of lead synthesis can be categorized in terms of reconstruction algorithms and lead configuration. The lead configuration for ECG synthesis can be divided into two groups: use of subsets of 12-lead ECG [28,29,30,31] and use of Frank VCG leads [32,33,34,35,36,37,38,39,40]. A common assumption in previous works was that the heart-torso electrical system is linear and quasi-static, which allows for the use of linear transformation to derive the 3-lead VCG from reduced-lead set of the 12-lead ECG. These can either be patient-specific or generic transformation of which the former is learned using data from a single patient, while the latter requires data from a group of patients. Previous studies have shown the possibility to derive the 12-lead ECG from the three Frank XYZ leads through Dower transformation [34] and vice versa through the inverse Dower transformation [35]. Similarly, Kors et al. [36] derived the transformation matrix using the regression analysis method. In [37], Dawson et al. derived the linear affine transformation between 3-lead VCG and 12-lead ECG, which achieved higher accuracy than Kors and inverse Dower transformation. Another strategy can be seen in [31,32], where nonlinear methods such as ANN were used to synthesize the 12-lead ECG and 3-lead VCG from leads I, II, and $V_{2}$. It was found that nonlinear transformation are appropriate for ECG data with diversity resulting from variation in individuals and measurement positions. A weakness for majority of the reviewed methods is that they only exploited the inter-lead correlation between spatially aligned samples of the lead signals. It is important to note that, in addition to spatially correlated information in different leads, temporally correlated information can also be found between different waves within a single lead. System design approaches that consider both intra- and inter-lead correlation are expected to provide better solutions to the VCG synthesis problem. This task can be accomplished by using RNN [41] as it can use the learning capabilities of ANN and could further improve it by representing the spatio-temporal correlations between the lead signals. In this work, we proposed a patient-specific transformation for VCG synthesis by applying a long short-term memory (LSTM) network [42] with sliding window approach.

This study focuses on two issues: synthesis of 3-lead VCG and extraction of VCG features, to develop an MI classifier that is suitable for wearable devices with only a single lead recording. The first part of this study focuses on developing a method of VCG reconstruction from lead I ECG using a LSTM network to exploit both intra- and inter-lead correlations of ECG signals. The second part of this study develops a novel spline framework for parametrically representing the derived Frank lead signals. After extracting features by the spline approximation, a classifier is used for the classification of healthy and 11 types of MI.

## 2. Methods

This study proposes a new method for automatic MI classification using the single-lead derived VCG. As shown in Figure 2, the proposed method consists of four stages, i.e., preprocessing, VCG synthesis, feature extraction, and classification. The raw ECG signals are preprocessed to remove various kinds of noise associated with them. Next, a patient-specific reconstruction method is used to synthesize the 3-lead VCG from lead I ECG. In the feature extraction stage, the clinically significant features are extracted from three derived Frank leads that quantify the VCG abnormalities due to MI. Later in the classification stage, the most likely ECG class has to be predicted from the analysis of the feature data.

### 2.1. Preprocessing

The raw ECG signal is typically contaminated by high-frequency noises caused by power-line interference, electromyographic noises due to muscle activity, motion artifacts caused by patient’s movements, and radio frequency noises from other equipments. Moreover, baseline wander is low-frequency (0–0.5 Hz) interference in the ECG signal caused by respiration, body movement and changes in electrode impedance. These noises degrade the quality of ECG signals and introduce ambiguity in the MI classification. Hence, the preprocessing is generally performed to to remove various types of noises associated with the input signal. The guidelines for the standardization and interpretation of ECG, published by the American Heart Association [43], advise using a cutoff frequency of 0.05 Hz for the high-pass filter and 150 Hz for the low-pass filter in adults. Thus, in this study, the raw ECG signal is down-sampled to 500 Hz and then filtered using a band-pass filter with a bandwidth between 0.5 and 150 Hz to remove noise and baseline wander. A similar approach has been used in several other studies [16,44].

### 2.2. VCG Synthesis

Synthesis of 3-lead VCG from reduced-lead set of 12-lead ECG [32,35,36,37,38,39,40] has been investigated in the past to satisfy the need for more wearing comfort and ambulatory situations. Most methods [35,36,37,38,39,40] are based on linear transformation and the differences between them are in coefficients of transformation matrices. In [32], Vozda et al. used nonlinear methods such as ANN to synthesize the 3-lead VCG from quasi-orthogonal leads I, II, and $V_{2}$. Most current approaches to VCG synthesis focus on the inter-lead correlation, with less emphasis placed on the intra-lead correlation. The ECG signals from leads I, $V_{x},V_{y}$, and $V_{z}$ are shown in Figure 3. It can be observed that, in addition to spatially correlated information in different leads, temporally correlated information can also be found between different waves within a single lead. The lead signals are narrow angle projections of the same electric heart vector and hence correlations can be found among the signals of various leads. Moreover, the cardiac cycle is quasi-periodic in nature and hence intra-correlations are evident between different characteristic waves. A model which can simultaneously learn the intra- and inter-lead correlations of ECG signals is expected to further improve the reconstruction accuracy. This is because synthesizing a VCG lead essentially involves estimating morphology of the waveform and timings of the characteristic waves. The morphology information holds significant similarity within a lead and hence it can be obtained by exploiting the intra-lead correlation. Similarly, inter-lead correlation can be used to derive the temporal information because timings of the characteristic waves are highly correlated between synchronously recorded leads. This can be achieved by using RNN [41] based models as they can combine information from the present and previous inputs to decide the present output. Recognizing this, we propose a patient-specific VCG synthesis method based on a sliding-window approach together with LSTM network [42]. At the model estimation stage, the LSTM parameters were estimated for each individual by considering the lead I ECG as input and Frank XYZ leads as output of the model.

The LSTM network is commonly used for time series modeling because it solves the gradient vanishing problem by incorporating gate units and memory cells. In an LSTM, the error information is preserved and is back-propagated through the layers which essentially helps the model to learn over a large number of time-steps. The system architecture of the proposed VCG synthesizer is shown in Figure 4. The system starts by applying a sliding window which spreads a segment of currently available lead I data across the input neurons of LSTM. Then, we use an LSTM network to reconstruct three Frank leads by applying a transformation based on the data series in each window. Let $x_{t}$ denote the lead I ECG data at time *t* and let $y_{t}^{\left(1\right)},y_{t}^{\left(2\right)},y_{t}^{\left(3\right)}$ denote the Frank X, Y, Z lead data, respectively. For a sliding window of size *L*, suppose that the pair $\left(s_{t},y_{t}\right)$ at time *t* contains the data series $s_{t}=\left\{x_{t-L+1},x_{t-L+2},\ldots ,x_{t}\right\}$ and its corresponding target output $y_{t}=\left\{y_{t}^{\left(1\right)},y_{t}^{\left(2\right)},y_{t}^{\left(3\right)}\right\}$. Given a set of *T* training data pairs $\left\{\left(s_{t},y_{t}\right),t=1,2,\ldots ,T\right\}$, learning the derived VCG model consists of finding a function *F* which minimizes the mean square error between the original signal $y_{t}$ and its reconstructed signal ${\hat{y}}_{t}=F\left(s_{t}\right)$. Proceeding in this way, we transform the VCG synthesis problem into a supervised learning problem.

An LSTM model has the units composed of a memory cell, an input gate, an output gate and a forget gate. The structure of the LSTM unit is shown in Figure 5. An LSTM unit computes a mapping from the input $x_{t}$ to output $y_{t}$ by calculating the network unit activations using eq:Equation 1eq:Equation 5 iteratively from t=1 to *T*.
(1)$$f_{t}=\sigma \left(W_{f}x_{t}+U_{f}h_{t-1}+b_{f}\right)$$
(2)$$i_{t}=\sigma \left(W_{i}x_{t}+U_{i}h_{t-1}+b_{i}\right)$$
(3)$$c_{t}=f_{t}\circ c_{t-1}+i_{t}\circ \tanh\left(W_{c}x_{t}+U_{c}h_{t-1}+b_{c}\right)$$
(4)$$o_{t}=\sigma \left(W_{o}x_{t}+U_{o}h_{t-1}+b_{o}\right)$$
(5)$$y_{t}=h_{t}=o_{t}\circ \tanh\left(c_{t}\right)$$
where *W*, *U*, and *b* denote the weight matrices and bias vectors which need to be learned during training. The operator ∘ denotes the element-wise product and σ is the sigmoid function. $c_{t}$ is the cell state, $h_{t}$ is the hidden state, and $f_{t}$, $i_{t}$, $o_{t}$ represent the forget gate, input gate and output gate, respectively. A series of experiments were performed to optimize the LSTM topology used for the VCG synthesizer. The networks with 1, 2, and 3 hidden layers and different number of neurons in hidden layers were tested. It was found that a network with two hidden layers and 30 neurons in each hidden layer achieved the best accuracy of transformation. The LSTM was trained using backpropagation through time (BPTT) algorithm [45], combined with the stochastic gradient descent algorithm. Adam optimizer was used in the model fine-tuning phase to further determine the LSTM parameters. Selected through iterative experiments, a time-step of 1, a mini-batch size of 128, and an epoch number of 300 were used to minimize the mean square error of the VCG synthesizer.

### 2.3. Feature Extraction

In the feature extraction stage, Frank XYZ leads of the derived VCG were individually processed in the following steps. First, we detect the R peak in each QRS complex using the Pan-Tompkins algorithm [20] and split the signals into heartbeat segments between two neighboring R peaks. Since the heartbeats may have different lengths, each heartbeat is period normalized to a fixed length of 400 samples via cubic spline interpolation. This choice was based on the observation that the average heartbeat length is about 0.8 s, which corresponds to 400 samples for a sampling frequency of 500 Hz. To make different lead signals comparable to each other, the min-max normalization was applied to scale both the amplitude and time in the range of [0,1], as described in [46]. For the *i*-th heartbeat with length $N_{i}$, let $\alpha_{i}=N_{i}/400$ denote the time scaling factor and let $\beta_{i}^{\left(1\right)},\beta_{i}^{\left(2\right)},\beta_{i}^{\left(3\right)}$ denote the amplitude scaling factor of Frank X, Y, Z lead, respectively. Once the heartbeats have been segmented and normalized, spline curve fitting [23] is applied to the entire heartbeat to model all of the characteristic waves and fitted coefficients are used as VCG representing features. Two advantages are provided. First, by using the entire beat, the method not only obviates the need for ECG wave delineation but also provides better representation of all regions of ECG beats for MI classification. Second, splines provide an efficient and accurate representation of VCG signals with semiperiodic patterns. VCG signals are a special type of semiperiodic signal which exhibits different degrees of smoothness in different intervals. Such signals require a higher-order polynomial to fit, leading to severe oscillations of the fitted curve which cause the overfitting problem [25]. To address this problem, we develop a framework for an efficient representation of Frank lead signals using splines.

Splines are piecewise polynomial approximations of a signal defined by constraint points on each piecewise segment known as knots. Since VCG signal has numerous clinically relevant turning points, the spline represented as a linear combination of *p*-degree B-spline basis function has been chosen as the approximation function. The knot vector ${\left\{\zeta_{j}\right\}}_{0}^{m}=\left\{\zeta_{j},0\leq j\leq m\right\}$ is a non-decreasing sequence, where the first (p+1) knots are all equal to 0.0025 and the last (p+1) knots are all equal to 1. The knots from $\zeta_{p+1}$ to $\zeta_{m-p-1}$ correspond to interior knots which are generated via the knot averages [25] according to Equation (Equation 6).
(6)$$\begin{array}{c} \zeta_{k}=\frac{\left(\tau_{k+1}+\tau_{k+2}+\cdots +\tau_{k+p}\right)}{p},\;p+1\leq k\leq m-p-1 \end{array}$$
where $\left\{\tau_{p+1},\tau_{p+2},\ldots ,\tau_{m}\right\}$ is an arithmetic sequence with the first term $\tau_{p+1}=0.0025$ and the last term $\tau_{m}=1$. The spline curve approximation can be expressed in the form of Equation (Equation 7).
(7)$$\begin{array}{c} u\left(t\right)=\sum_{i=0}^{n}a_{i}B_{i,p}\left(t\right), \end{array}$$
where n=m−p−1 and $a_{i}$ represents the *i*-th B-spline coefficient. $B_{i,p}\left(t\right)$ denotes the *i*-th *p*-degree B-spline basis function which is computed recursively [25] using Equations (Equation 8) and (Equation 9).
(8)$$\begin{array}{c} \begin{array}{c} B_{i,0}\left(t\right)=\left\{\begin{array}{cc} 1, & \zeta_{i}\leq t\leq \zeta_{i+1}\\ 0, & \mathrm{otherwise} \end{array}\right. \end{array} \end{array}$$
(9)$$\begin{array}{c} B_{i,j}\left(t\right)=\frac{t-\zeta_{i}}{\zeta_{i+j}-\zeta_{i}}B_{i,j-1}\left(t\right)+\frac{\zeta_{i+j+1}-t}{\zeta_{i+j+1}-\zeta_{i}}B_{i+1,j-1}\left(t\right) \end{array}$$

The vector of coefficients $\left\{a_{i},0\leq i\leq n\right\}$ is calculated by using the least square spline approximation. Generally, the B-spline approximation of VCG signal yielded better performance with an increase in the value of *n*. Figure 6 shows the original heartbeat and the spline fitting curve with n=23 using one MI sample and one healthy sample. Experimentally, it was found that the use of n=15 gives a good trade-off between computational efficiency and the quality of fit. Each normalized heartbeat is transformed into 16 features $\left\{a_{0},a_{1},\ldots ,a_{15}\right\}$, and three VCG leads during the time of a given heartbeat have 48 features. Together with the time scaling factor $\alpha_{i}$ and amplitude scaling factors $\left\{\beta_{i}^{\left(1\right)},\beta_{i}^{\left(2\right)},\beta_{i}^{\left(3\right)}\right\}$, the complete heartbeat of 3-lead VCG is transformed as a 52-dimensional feature vector.

### 2.4. Classification

The system performance of MI classification depends critically on the underlying classifier, which builds a model of how to best predict which class a test ECG beat belongs. In this study, a classifier based on multilayer perceptron network (MLP) is used for classification into 12 classes of ECG beats. The MLP is a class of feedforward ANN model and widely used in many fields, such as object recognition, pattern classification, and biological data analysis. Among the reasons for this popularity are its nonlinearity, parallelism, learning and generalization capabilities [47]. A MLP is a network composed of parallel layers of neurons. In building MI classifiers, the input layer receives spline-fitted features from the derived VCG, and the output layer provides the predicted ECG classes. The relations between the input and output layers are expressed through the weights and biases of the hidden layer. All of the weights were initialized to small random numbers and then subjected to incremental changes by the error backpropagation algorithm based on the cross-entropy loss function [48]. To optimize the classifier design, we tested the MLP with 1, 2, and 3 hidden layers and the number of neurons in each hidden layer was tuned by a grid search from 50 to 500 in steps of 25. Based on the results, we chose the MLP network with 52 input nodes (one for each spline-fitted feature), 12 output nodes (one for each ECG class) and two hidden layers which had 300 and 275 nodes, respectively. To describe the intensity of neural firing, a neuron output was generally obtained by applying an activation function to the weighted sum of its inputs. Due to its ability to enable fast training, the rectified linear unit (ReLU) activation function [47] was used for the hidden layer. However, the ReLU nonlinearity is not applicable for the activation of the present output-layer neurons because their respective output values represent a categorical probability distribution. With this consideration, we applied the softmax function for the output layer to generate values which are in the unit interval and summed to one. Since MI diagnosis involves the simultaneous discrimination of several ECG classes, we considered the one-hot encoding [49] scheme for solving the categorical data classification problem. Specifically, the MLP outputs are represented as binary vectors, each vector consists of 0 s in all cells with the exception of a single 1 in an entry corresponding to the most likely class.

## 3. Evaluation Parameters

In this study, ECG records were taken from the Physikalisch-TechnischeBundesanstalt (PTB) [50] diagnostic database. The PTB database consists of 549 ECG records from 290 subjects and each record contains 12 ECG leads and 3 Frank VCG leads. From the database, a total of 26,080 heartbeats from 52 healthy subjects and 143 MI patients were included in the analysis. Table 1 shows the number of heartbeats for each type of MI and healthy subjects in this study. These data were further divided into 12 classes of ECG beats: anterior (AMI), anterior-lateral (ALMI), anterior-septal (ASMI), anterior-septal-lateral (ASLMI), inferior (IMI), inferior-lateral (ILMI), inferior-posterior (IPMI), inferior-posterior-lateral (IPLMI), lateral (LMI), posterior (PMI), posterior-lateral (PLMI), and healthy control (HC).

Root-mean-square-error (RMSE) and correlation coefficient (CC) were chosen to test the accuracy of derived VCG by the individual methods in relation to the measured VCG. RMSE measures the similarity of two recordings and it is defined as Equation (Equation 10), where *V* is the original value of the measured VCG, $\hat{V}$ is the value of the derived VCG, and *N* is the number of samples. Instead, CC is a statistic that measures the correlation between two recordings, which is defined in Equation (Equation 11).
(10)$$RMSE=\sqrt{\frac{1}{N}\sum_{i=1}^{N}{\left(V_{i}-{\hat{V}}_{i}\right)}^{2}}$$
(11)$$CC=\frac{\sum_{i=1}^{N}V_{i}\cdot {\hat{V}}_{i}}{\sqrt{\sum_{i=1}^{N}V_{i}^{2}\sum_{i=1}^{N}{\hat{V}}_{i}^{2}}}$$

To test the feasibility of the proposed MI classifiers, the performance analysis is based on the accuracy (ACC), sensitivity (SEN), and specificity (SPE) represented in the form of confusion matrix [51]. These performance metrics are related to the number of true positives (TP), true negatives (TN), false positives (FP), and false negatives (FN). The accuracy is the proportion of correctly classified samples to the total number of samples, and it is defined as Equation (Equation 12). Sensitivity, defined in Equation (Equation 13), measures the proportion of positives that are correctly identified. Instead, specificity measures the proportion of negatives that are correctly identified and defined as Equation (Equation 14).
(12)$$ACC=\frac{TP+TN}{TP+FP+TN+FN}$$
(13)$$SEN=\frac{TP}{TP+FN}$$
(14)$$SPE=\frac{TN}{FP+TN}$$

## 4. Results

Computer simulations were conducted to evaluate the validity of the proposed method in differentiating 11 types of MI and healthy subjects. A preliminary experiment was first conducted to examine the performance dependence of VCG synthesis on the sliding window size *L* employed in constructing the LSTM models. In this experiment, ECG recordings from 20 HC subjects and 20 MI patients were used. Table 2 presents the RMSE and CC between measured and derived Frank XYZ leads. Based on the results, we empirically chose L=150 in the sequel.

We next compare the reconstruction performance of using MLP [32] and LSTM for learning the derived VCG models. All the experiments were based on the evaluation of RMSE and CC and experimental results were obtained by five-fold cross-validation. ECG recordings from 52 HC subjects are denoted as dataset DS1, and ECG recordings from 143 MI patients are denoted as dataset DS2. For comparison purposes, the MLP consists of one input layer with 150 neurons, one output layer with three neurons, two hidden layers and 150 neurons per hidden layer. The results of VCG synthesis by five-fold cross-validation are presented in Table 3. The results clearly demonstrate that the LSTM is preferred to MLP for use in constructing the VCG synthesizer because the LSTM can exploit both intra- and inter-lead correlations of ECG signals. Further analysis indicates that the average CC of three Frank leads using LSTM were 0.9943 and 0.9807 for dataset DS1 and DS2, respectively, suggesting that the MI patient data was less accurately reconstructed than HC subjects. Visual inspection of the reconstructed signals showed that the derived VCG signals were not significantly different from the measured signals. A typical example for measured and derived Frank XYZ leads is depicted in Figure 7.

Next, we assess the performance of MLP classifiers for the classification of normal and 11 MI classes. One problem with the PTB database is the high imbalance between the number of heartbeats belonging to each ECG class. Training an MLP classifier with unbalanced data usually leads to a certain bias towards the majority class. Recognizing this, we applied the Synthetic Minority Over-sampling Technique (SMOTE) [52] before starting the training process. Moreover, we used 5-fold cross-validation technique to train and test the MLP classifiers. We began testing the MLP classifiers for the situation where MI classes were identified solely by means of single-lead feature data. The classification performance for each ECG class is summarized in Table 4. Simulation results indicated that using lead I yielded an overall accuracy of 50.72%, suggesting that it cannot provide sufficient cues for reliable classification. To elaborate further, we show in Table 5 the confusion matrix of all the 12 classes for lead I ECG beats. It was found that the notably low classification accuracy can be attributed to the high confusions made across anterior MI group (AMI, ALMI, ASMI, ASLMI) and inferior MI group (IMI, ILMI, IPMI, IPLMI). For instance, 14.71% of the ECG beats notated in ASMI were classified as representing IMI and 8.84% of the IMI beats were classified as being ASMI. Furthermore, results indicate that derived Frank leads ${\hat{V}}_{y}$ and ${\hat{V}}_{z}$ are preferred to lead I ECG for use in constructing the MI classifier. Notably, the use of lead ${\hat{V}}_{z}$ yielded an overall accuracy of 82.09%, compared with 50.72% for lead I and 81.45% for ${\hat{V}}_{y}$. The confusion matrices obtained using derived Frank lead ${\hat{V}}_{y}$ and ${\hat{V}}_{z}$ are shown in Table 6 and Table 7, respectively. Further analysis indicates that anterior and inferior MI groups are dominant in the MI groups that benefited the most from exploitation of derived VCG leads. In case of inferior MI group, the average sensitivity has increased from 56.9% in lead I to 87.1% in lead ${\hat{V}}_{y}$. Similarly in case of anterior MI group, average sensitivity obtained for lead I and ${\hat{V}}_{z}$ is 60.25% and 88.9%, respectively. We speculate that this might be attributed to the difference in closeness between Frank leads and 12 ECG leads. Support for such a speculation can be found in [39], where the authors showed that Frank lead $V_{y}$ is most likely associated with inferior leads (II, III, aVF), and Frank lead $V_{z}$ is closest to subset of anteroseptal leads $\left(V_{1},V_{2},V_{3}\right)$. We can also see from Figure 1 that leads $V_{1},V_{2}$ and $V_{3}$ are located near the negative Z-axis in the sagittal plane. Similarly, it can be found that leads II, III and aVF are oriented along the Y-axis.

Next, we examine whether combining multiple derived Frank leads would improve the classification performance. Table 8 shows the MLP classifier results for healthy and 11 types of MI ECG beats obtained using various lead configurations. Our proposed method yielded the best performance with an overall accuracy of 99.15%, sensitivity of 99.16% and specificity of 99.92% in MI classification, by using 52 features obtained from the derived Frank XYZ leads. The results also indicate that the ability of derived VCG to correctly identify the MI classes is almost identical to that of measured VCG. Table 9 shows the confusion matrix of all classes obtained using MLP classifier on the derived Frank XYZ leads. A comparison between Table 5 and Table 9 indicates that the improvement can be seen in the following areas. First, the derived VCG can reduce a significant portion of confusions across anterior and inferior MI groups. For instance, only 0.15% of ECG beats notated in ASMI were misclassified as representing IMI, the corresponding value for lead I being 14.71%. Second, the derived VCG significantly increased the sensitivity of healthy subjects to 99.68%, compared with 58.78% for lead I, 78.99% for ${\hat{V}}_{y}$, and 80.21% for ${\hat{V}}_{z}$. The results clearly demonstrate that MI classification by computational means is significantly improved when clinically significant features relating to the derived VCG are taken into account.

## 5. Discussion

In recent years, numerous approaches were proposed to identify various types of MI from ECG records. The numbers of ECG leads and MI classes are important factors correlated with diagnosis efficiency, and should be noted when comparing their relative performances. Table 10 summarizes the studies employing different techniques in MI classification with the same PTB database. Arif et al. [10] used 12 lead ECG signal and time domain features such as T-wave amplitude, Q-wave and ST-level elevation, reporting overall accuracy of 98.8% on ten different MI classes with a KNN classifier. Alternatively, Noorian et al. [11] used ANN classifier and wavelet coefficients as features extracted from the derived VCG. Acharya et al. [12] have evaluated ten MI classes with 12 types of nonlinear features based on wavelet transform. They obtained an accuracy of 98.74%, sensitivity of 99.55%, and specificity of 99.16% by only using lead V3 ECG signal. Lui et al. [13] combined the power of CNN and RNN, and achieved 92.4% sensitivity and 97.7% specificity for classification of MI as well as other cardiovascular diseases. Baloglu et al. [14] proposed an end-to-end approach based on deep CNN and reported an overall accuracy of 99.78% by using 12 lead ECG signal for classification into 11 types of ECG beats. In [15], a multi-lead attention mechanism integrated with CNN and bidirectional gated recurrent unit was applied for MI classification based on six classes of 12-lead ECG records, namely HC, AMI, ALMI, ASMI, IMI, and ILMI. Towards addressing the challenges in identifying MIs using wearable devices, our work, as well as some earlier studies [12,13], was focused on single-lead rather than 12-lead exploration. Results reported in this paper are generally better than those of MI classifiers in the literature, with its performance only slightly lower than that of [14]. However, our proposed method applies single-lead derived VCG for classification into 12 types of ECG beats, in which ASLMI with larger necrotic area is ignored in [14]. Overall, the proposed method obtained an accuracy of 99.15%, sensitivity of 99.16% and specificity of 99.92%. With this performance, our proposed model has the potential to provide an early and accurate diagnosis of MI in wearable ECG monitoring devices.

## 6. Conclusions

This paper proposed a new method for automatic MI classification using single-lead derived VCG. We first emphasized the importance of exploiting both intra-lead and inter-lead correlation for learning the derived VCG models. This task was accomplished by using a patient-specific transformation based on LSTM network with sliding window approach. Performance is further enhanced by using B-spline curve fitting to extract clinically significant features from the three derived Frank leads. After feature extraction, a classifier based on MLP network is used for classification into 12 types of ECG beats. Combined performance from 52 healthy subjects and 143 MI patients demonstrate the validity of the proposed MI classification system with an accuracy of 99.15%, sensitivity of 99.16% and specificity of 99.92%.

## Figures and Tables

**Figure 1 sensors-20-07246-f001:**
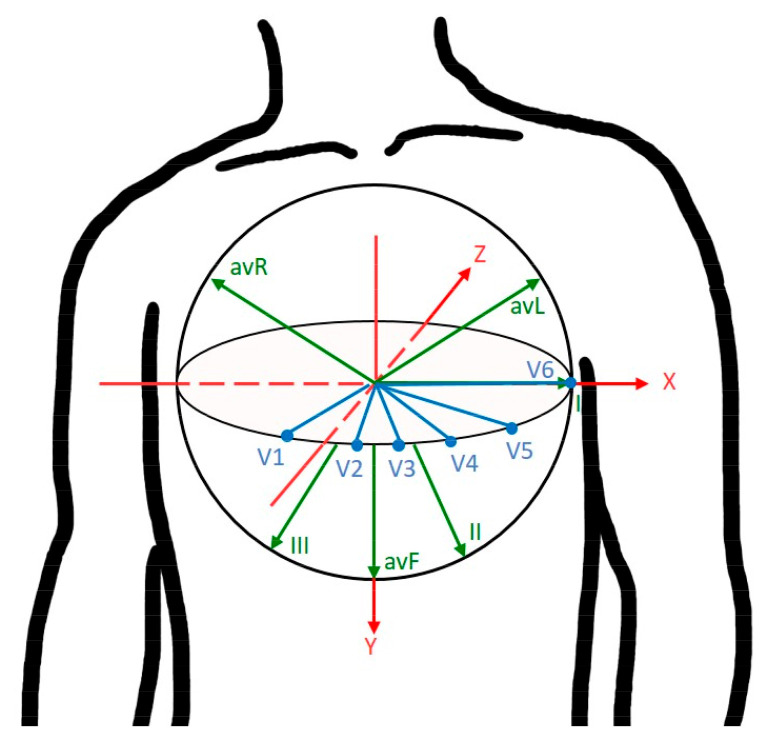
The three-dimensional view of 12 ECG leads on the *xyz*-coordinate axis system.

**Figure 2 sensors-20-07246-f002:**
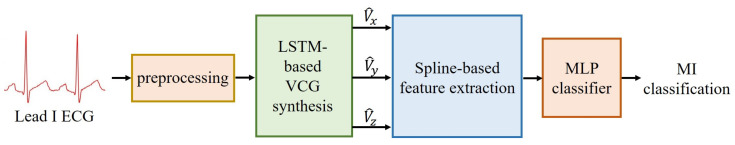
Block diagram of the proposed MI classification system.

**Figure 3 sensors-20-07246-f003:**
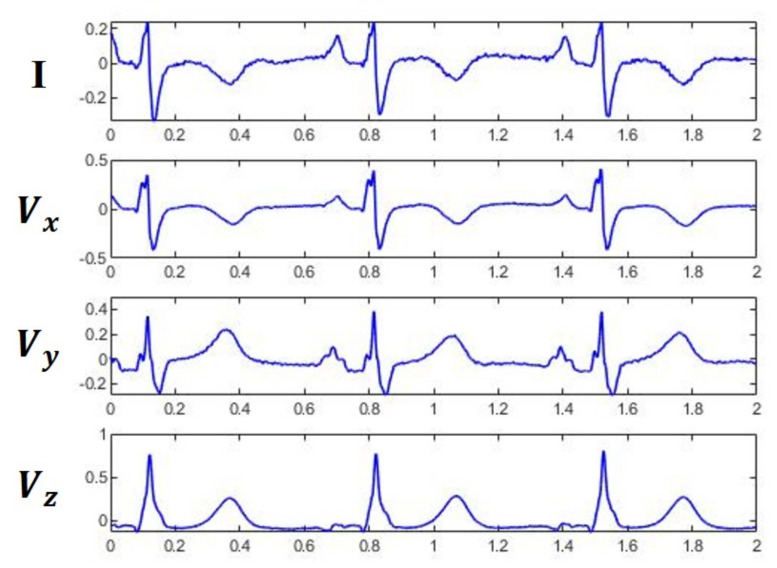
ECG waveforms of measured lead I and Frank XYZ leads.

**Figure 4 sensors-20-07246-f004:**
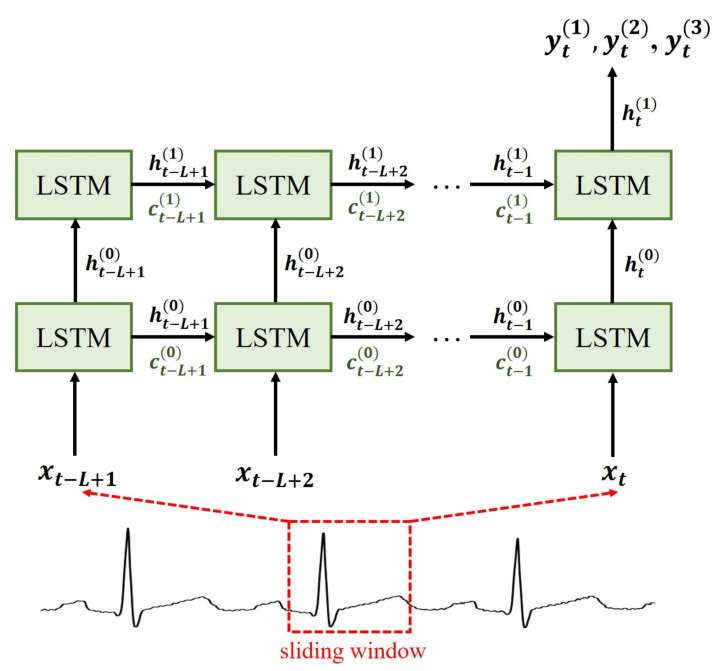
System architecture of the proposed VCG synthesizer.

**Figure 5 sensors-20-07246-f005:**
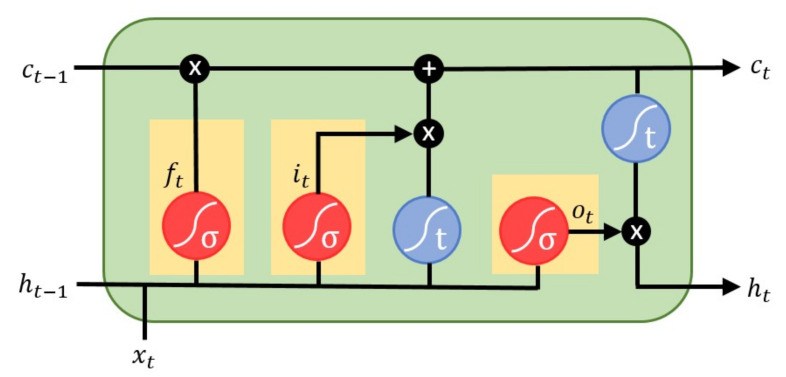
Structure of the LSTM unit.

**Figure 6 sensors-20-07246-f006:**
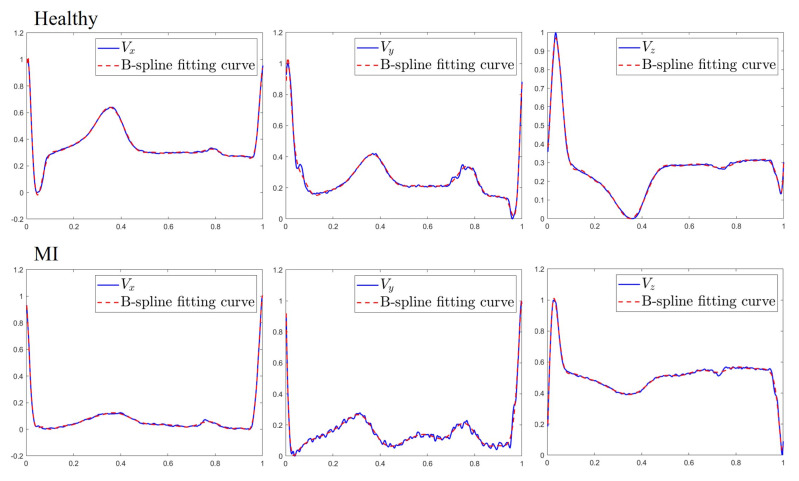
Comparison between original heartbeats (**blue**) and fitting curves (**red**) for healthy and MI subjects.

**Figure 7 sensors-20-07246-f007:**
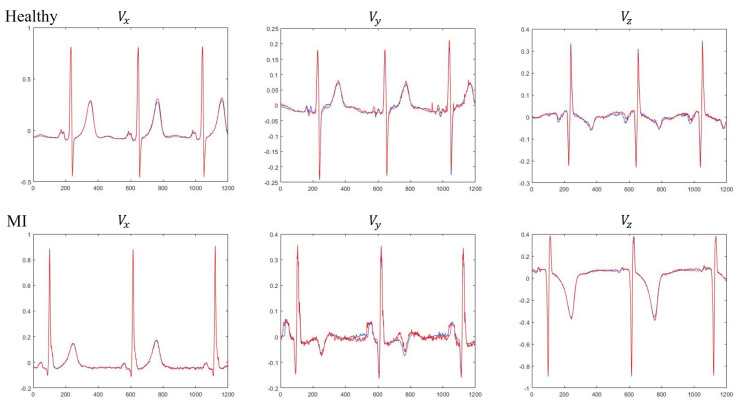
Comparison between measured (**blue**) and derived (**red**) Franks leads for healthy and MI subjects.

**Table 1 sensors-20-07246-t001:** Number of beats for different types of MI and healthy subjects in this study.

Class	Number of Beats
**Anterior (AMI)**	2800
**Anterior-Lateral (ALMI)**	2534
**Anterior-Septal (ASMI)**	4114
**Anterior-Septal-Lateral (ASLMI)**	134
**Inferior (IMI)**	4569
**Inferior-Lateral (ILMI)**	3143
**Inferior-Posterior (IPMI)**	336
**Inferior-Posterior-Lateral (IPLMI)**	1063
**Lateral (LMI)**	159
**Posterior (PMI)**	137
**Posterior-Lateral (PLMI)**	288
**Healthy Control (HC)**	6803

**Table 2 sensors-20-07246-t002:** RMSE and CC between measured and derived Frank XYZ leads.

Window Size	Performance	${\hat{V}}_{x}$	${\hat{V}}_{y}$	${\hat{V}}_{z}$
50	CC	0.9939	0.9789	0.9890
	RMSE	13.2215	16.9526	14.2964
100	CC	0.9956	0.9843	0.9933
	RMSE	11.6335	14.6710	10.8368
150	CC	0.9963	0.9862	0.9940
	RMSE	11.0374	14.0393	10.0226
200	CC	0.9962	0.9855	0.9939
	RMSE	11.2348	14.3675	10.2153

**Table 3 sensors-20-07246-t003:** Five-fold cross validation and average CC and RMSE between measured and derived Frank leads for the MLP and LSTM models.

Folds	Performance	Model	DS1			DS2		
			${\hat{V}}_{x}$	${\hat{V}}_{y}$	${\hat{V}}_{z}$	${\hat{V}}_{x}$	${\hat{V}}_{y}$	${\hat{V}}_{z}$
Fold 1	CC	MLP	0.9947	0.9732	0.9830	0.9815	0.9396	0.9716
		LSTM	0.9977	0.9881	0.9941	0.9909	0.9680	0.9885
	RMSE	MLP	17.2434	21.4437	21.4656	24.4987	28.9382	28.5079
		LSTM	11.4825	14.3947	10.2566	16.9540	20.7532	16.9667
Fold 2	CC	MLP	0.9949	0.9745	0.9835	0.9784	0.9388	0.9718
		LSTM	0.9981	0.9898	0.9963	0.9895	0.9686	0.9879
	RMSE	MLP	17.3081	20.4252	21.2486	24.3083	29.0842	28.3771
		LSTM	11.1590	12.6073	9.7404	15.9387	20.9914	17.4463
Fold 3	CC	MLP	0.9946	0.9719	0.9835	0.9773	0.9398	0.9720
		LSTM	0.9979	0.9878	0.9959	0.9864	0.9659	0.9875
	RMSE	MLP	17.6641	20.9450	21.6205	24.2560	29.7097	28.2240
		LSTM	11.1632	13.1473	10.0268	16.9523	21.6060	16.9857
Fold 4	CC	MLP	0.9952	0.9746	0.9843	0.9798	0.9390	0.9716
		LSTM	0.9986	0.9901	0.9966	0.9880	0.9695	0.9883
	RMSE	MLP	16.6822	20.1005	20.9287	24.3547	29.1888	28.4182
		LSTM	10.0381	12.0188	9.3277	16.0953	20.4111	16.7269
Fold 5	CC	MLP	0.9950	0.9758	0.9800	0.9784	0.9349	0.9692
		LSTM	0.9985	0.9922	0.9929	0.9847	0.9618	0.9844
	RMSE	MLP	16.9020	20.3295	21.7694	25.3288	30.9232	29.7329
		LSTM	10.0791	11.5885	9.6506	17.9217	22.7715	18.0954
Mean	CC	MLP	0.9949	0.9740	0.9829	0.9791	0.9384	0.9712
		LSTM	0.9982	0.9896	0.9952	0.9879	0.9668	0.9873
	RMSE	MLP	17.1600	20.6488	21.4066	24.5493	29.5688	28.6520
		LSTM	10.7844	12.7513	9.8004	16.7724	21.3066	17.2442

**Table 4 sensors-20-07246-t004:** Classification results of MLP classifier with single-lead signal.

Classes	I			${\hat{V}}_{x}$			${\hat{V}}_{y}$			${\hat{V}}_{z}$		
	ACC(%)	SEN(%)	SPE(%)	ACC(%)	SEN(%)	SPE(%)	ACC(%)	SEN(%)	SPE(%)	ACC(%)	SEN(%)	SPE(%)
**AMI**	90.55	51.89	95.20	91.74	58.39	95.75	95.73	79.71	97.66	95.76	79.57	97.70
**ALMI**	91.33	51.10	95.66	93.01	57.54	96.83	96.68	87.10	97.72	97.58	88.60	98.54
**ASMI**	85.28	37.99	94.13	88.35	56.73	94.27	93.60	75.62	96.96	95.41	87.29	96.93
**ASLMI**	99.40	100.00	99.40	99.75	100.00	99.75	99.89	98.51	99.90	99.99	100.00	99.99
**IMI**	80.87	38.26	89.93	81.94	33.44	92.24	93.49	81.02	96.14	92.08	73.95	95.93
**ILMI**	86.19	54.82	90.49	87.81	61.47	91.42	95.82	84.16	97.41	94.69	83.33	96.25
**IPMI**	99.09	92.26	99.18	99.36	89.88	99.48	99.78	94.94	99.84	99.70	93.15	99.78
**IPLMI**	93.95	42.24	96.15	93.17	35.18	95.63	98.76	88.33	99.20	97.67	81.84	98.34
**LMI**	98.12	98.11	98.12	99.00	99.37	99.00	99.69	100.00	99.68	99.83	98.11	99.84
**PMI**	99.24	100.00	99.23	98.49	100.00	98.48	99.51	100.00	99.51	99.89	100.00	99.89
**PLMI**	96.92	90.63	97.00	97.25	87.85	97.36	99.65	98.96	99.65	99.76	96.88	99.79
**HC**	80.49	58.78	88.15	85.20	69.98	90.57	90.30	78.99	94.29	91.83	80.21	95.93

**Table 5 sensors-20-07246-t005:** Confusion matrix for MI classification using measured lead I ECG.

Notated	Predicted												Total	ACC(%)	SEN(%)	SPE(%)
	AMI	ALMI	ASMI	ASLMI	IMI	ILMI	IPMI	IPLMI	LMI	PMI	PLMI	Norm				
**AMI**	1453	255	150	0	335	228	21	97	44	2	39	176	2800	90.55	51.89	95.2
**ALMI**	154	1295	219	54	228	141	37	57	2	128	129	90	2534	91.33	51.1	95.66
**ASMI**	273	217	1563	25	605	555	41	101	129	10	125	470	4114	85.28	37.99	94.13
**ASLMI**	0	0	0	134	0	0	0	0	0	0	0	0	134	99.4	100	99.4
**IMI**	219	279	404	65	1748	372	72	144	51	35	177	1003	4569	80.87	38.26	89.93
**ILMI**	173	82	113	0	281	1723	26	204	122	3	5	411	3143	86.19	54.82	90.49
**IPMI**	5	8	3	0	9	0	310	0	0	0	0	1	336	99.09	92.26	99.18
**IPLMI**	31	68	77	1	84	155	2	449	53	1	19	123	1063	93.95	42.24	96.15
**LMI**	2	0	0	0	0	1	0	0	156	0	0	0	159	98.12	98.11	98.12
**PMI**	0	0	0	0	0	0	0	0	0	137	0	0	137	99.24	100	99.23
**PLMI**	0	1	1	0	7	3	0	5	0	0	261	10	288	96.92	90.63	97
**HC**	260	111	322	11	618	726	12	356	87	20	281	3999	6803	80.49	58.78	88.15

**Table 6 sensors-20-07246-t006:** Confusion matrix for MI classification using derived Frank Y lead.

Notated	Predicted												Total	ACC(%)	SEN(%)	SPE(%)
	AMI	ALMI	ASMI	ASLMI	IMI	ILMI	IPMI	IPLMI	LMI	PMI	PLMI	Norm				
**AMI**	2232	44	113	0	137	55	4	32	5	32	4	142	2800	95.73	79.71	97.66
**ALMI**	35	2207	49	15	87	25	3	9	0	34	1	69	2534	96.68	87.1	97.72
**ASMI**	161	78	3111	9	176	103	12	25	0	0	36	403	4114	93.6	75.62	96.96
**ASLMI**	0	1	0	132	0	0	0	0	0	0	1	0	134	99.89	98.51	99.9
**IMI**	95	116	157	2	3702	136	6	72	12	2	38	231	4569	93.49	81.02	96.14
**ILMI**	8	15	50	1	130	2645	4	21	26	0	3	240	3143	95.82	84.16	97.41
**IPMI**	3	0	4	0	5	1	319	0	0	0	0	4	336	99.78	94.94	99.84
**IPLMI**	26	7	5	0	43	27	2	939	2	1	0	11	1063	98.76	88.33	99.2
**LMI**	0	0	0	0	0	0	0	0	159	0	0	0	159	99.69	100	99.68
**PMI**	0	0	0	0	0	0	0	0	0	137	0	0	137	99.51	100	99.51
**PLMI**	0	1	0	0	0	2	0	0	0	0	285	0	288	99.65	98.96	99.65
**HC**	217	276	289	0	252	244	10	40	37	58	6	5374	6803	90.3	78.99	94.29

**Table 7 sensors-20-07246-t007:** Confusion matrix for MI classification using derived Frank Z lead.

Notated	Predicted												Total	ACC(%)	SEN(%)	SPE(%)
	AMI	ALMI	ASMI	ASLMI	IMI	ILMI	IPMI	IPLMI	LMI	PMI	PLMI	Norm				
**AMI**	2228	74	272	1	76	24	8	14	5	6	9	83	2800	95.76	79.57	97.7
**ALMI**	45	2245	96	0	38	11	18	7	1	2	1	70	2534	97.58	88.6	98.54
**ASMI**	103	76	3591	2	105	87	4	13	0	0	0	133	4114	95.41	87.29	96.93
**ASLMI**	0	0	0	134	0	0	0	0	0	0	0	0	134	99.99	100	99.99
**IMI**	147	94	186	0	3379	254	18	80	14	3	28	366	4569	92.08	73.95	95.93
**ILMI**	18	17	35	0	202	2619	2	109	11	2	10	118	3143	94.69	83.33	96.25
**IPMI**	7	3	1	0	5	5	313	1	0	0	0	1	336	99.7	93.15	99.78
**IPLMI**	3	2	3	0	137	31	3	870	0	0	0	14	1063	97.67	81.84	98.34
**LMI**	0	0	0	0	2	0	0	1	156	0	0	0	159	99.83	98.11	99.84
**PMI**	0	0	0	0	0	0	0	0	0	137	0	0	137	99.89	100	99.89
**PLMI**	0	0	0	0	0	0	0	6	3	0	279	0	288	99.76	96.88	99.79
**HC**	212	77	82	0	311	448	3	184	8	15	6	5457	6803	91.83	80.21	95.93

**Table 8 sensors-20-07246-t008:** Classification results of MLP classifier with various lead configurations.

Leads	ACC(%)	SEN(%)	SPE(%)
I	50.72	68.01	95.22
${\hat{V}}_{x}$	57.54	70.82	95.90
${\hat{V}}_{y}$	81.45	88.95	98.17
${\hat{V}}_{z}$	82.09	88.58	98.24
${\hat{V}}_{x}+{\hat{V}}_{y}$	93.36	95.52	99.34
${\hat{V}}_{y}+{\hat{V}}_{z}$	96.99	97.74	99.70
${\hat{V}}_{x}+{\hat{V}}_{z}$	83.68	89.80	98.42
${\hat{V}}_{x}+{\hat{V}}_{y}+{\hat{V}}_{z}$	99.15	99.16	99.92
$V_{x}+V_{y}+V_{z}$	99.14	99.39	99.92

**Table 9 sensors-20-07246-t009:** Confusion matrix for MI classification using derived Frank XYZ leads.

Notated	Predicted												Total	ACC(%)	SEN(%)	SPE(%)
	AMI	ALMI	ASMI	ASLMI	IMI	ILMI	IPMI	IPLMI	LMI	PMI	PLMI	Norm				
**AMI**	2762	5	9	0	7	4	0	1	0	0	3	9	2800	99.72	98.64	99.85
**ALMI**	10	2504	6	0	2	2	2	0	1	0	0	7	2534	99.78	98.82	99.88
**ASMI**	9	12	4078	1	6	2	3	3	0	0	0	0	4114	99.73	99.12	99.85
**ASLMI**	0	0	0	134	0	0	0	0	0	0	0	0	134	100	100	100
**IMI**	5	3	4	0	4528	11	1	7	0	0	4	6	4569	99.7	99.1	99.83
**ILMI**	3	3	8	0	3	3119	1	1	1	0	1	3	3143	99.8	99.24	99.88
**IPMI**	3	0	0	0	3	0	329	0	0	0	0	1	336	99.94	97.92	99.97
**IPLMI**	2	3	3	0	6	8	0	1039	0	1	0	1	1063	99.85	97.74	99.94
**LMI**	0	0	0	0	0	0	0	0	159	0	0	0	159	99.99	100	99.99
**PMI**	0	0	0	0	0	0	0	0	0	137	0	0	137	99.99	100	99.99
**PLMI**	0	0	0	0	0	0	0	1	0	0	287	0	288	99.97	99.65	99.97
**HC**	2	2	4	0	10	1	1	1	0	1	0	6781	6803	99.81	99.68	99.86

**Table 10 sensors-20-07246-t010:** Comparison of this study with other studies using the PTB diagnostic database.

Ref	Leads	No. of Classes	ACC(%)	SEN(%)	SPE(%)
Arif et al. (2012) [10]	12 leads	11	98.80%	98.67%	98.71%
Noorian et al. (2014) [11]	12 leads	10	95.35%	99.09%	94.23%
Acharya et al. (2016) [12]	$V_{3}$	11	98.74%	99.55%	99.16%
Lui nad Chow (2018) [13]	I	4	95.25%	92.40%	97.70%
Baloglu et al. (2019) [14]	12 leads	11	99.78%	99.84%	99.98%
Fu et al. (2020) [15]	12 leads	6	99.11%	99.02%	99.10%
Proposed method	I	12	99.15%	99.16%	99.92%

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
