# Peer review of "Automatic Classification of Myocardial Infarction Using Spline Representation of Single-Lead Derived Vectorcardiography"

_sensors, 2020, doi:10.3390/s20247246_

Round 1

Reviewer 1 Report

The manuscript provided a smart method to detect MI using single-lead derived VCG. This will allow wearable devices to detect abnormalities within ECG and detect MI. The problem is significant and the approach seems sound. The authors may add some statistics and references regarding MI in Introduction. Consultation with a cardiologist is helpful to further analyze the data and find clinical applications. 

Reviewer 2 Report

This is an interesting paper which explores the use of machine learning to assess myocardial infraction using single-lead derived VCG. The article is of interest; however I do have several observations which mainly focus on the authors not providing sufficient justification for their approach.

  1. The title should be updated to reflect the methodology used in the paper. This includes avoiding the use of abbreviations.
  2. Line 16-17: Please provide a reference to support the statement, and please provide, if possible, the number of persons who die each year as a result of MI.
  3. The authors provide rationale for the method used in the paper within the introduction (ie. line 53-57).
  4. It would be helpful to end the introduction with a clear aim, objective or hypotheses.
  5. Pre-processing: The authors provide a pre-processing frame to improve the signal extract from raw ECG. It would be helpful for the authors to cite other studies which have used the same pre-processing or highlight if they are novel contributions.
  6. Line 118-119: If 3-lead VCG has been used in the past, please provide citations to the previous research.
  7. Line 126-127: Why would a model which can learn the intra and inter-lead correlations could improve accuracy?
  8. Did the authors consider comparing the performance of the current classification framework to other machine learning algorithms?
  9. For training the model, what were the specific parameters used? Did the authors follow a fine turning process?

Round 2

Reviewer 2 Report

The authors have addressed my concerns. 

This manuscript is a resubmission of an earlier submission. The following is a list of the peer review reports and author responses from that submission.